# Optimization of Smooth Functions with Noisy Observations: Local Minimax Rates

**Yining Wang, Sivaraman Balakrishnan, Aarti Singh**
Department of Machine Learning and Statistics
Carnegie Mellon University, Pittsburgh, PA, 15213, USA
{yiningwa,aarti}@cs.cmu.edu, siva@stat.cmu.edu

## Abstract

We consider the problem of *global optimization* of an unknown non-convex smooth function with noisy zeroth-order feedback. We propose a *local minimax* framework to study the fundamental difficulty of optimizing smooth functions with adaptive function evaluations. We show that for functions with fast growth around their global minima, carefully designed optimization algorithms can identify a near global minimizer with many fewer queries than worst-case global minimax theory predicts. For the special case of strongly convex and smooth functions, our implied convergence rates match the ones developed for zeroth-order *convex* optimization problems. On the other hand, we show that in the worst case no algorithm can converge faster than the minimax rate of estimating an unknown functions in $\ell_\infty$-norm. Finally, we show that non-adaptive algorithms, although optimal in a global minimax sense, do not attain the optimal local minimax rate.

## 1 Introduction

Global function optimization with stochastic (zeroth-order) query oracles is an important problem in optimization, machine learning and statistics. To optimize an unknown bounded function $f : \mathcal{X} \mapsto \mathbb{R}$ defined on a known compact $d$-dimensional domain $\mathcal{X} \subseteq \mathbb{R}^d$, the data analyst makes $n$ *active* queries $x_1, \ldots, x_n \in \mathcal{X}$ and observes

$$y_t = f(x_t) + w_t, \qquad w_t \stackrel{i.i.d.}{\sim} \mathcal{N}(0,1),^1 \ \ t = 1, \ldots, n. \tag{1}$$

The queries $x_1, \ldots, x_t$ are *active* in the sense that the selection of $x_t$ can depend on previous queries and their responses $x_1, y_1, \ldots, x_{t-1}, y_{t-1}$. After $n$ queries, an estimate $\widehat{x}_n \in \mathcal{X}$ is produced that approximately minimizes the unknown function $f$. Such "active query" models are relevant in a broad range of (noisy) global optimization applications, for instance in hyper-parameter tuning of machine learning algorithms [40] and sequential design in material synthesis experiments where the goal is to maximize strengths of the produced materials [35, 41]. Sec. 2.1 gives a rigorous formulation of the active query model and contrasts it with the classical passive query model.

The error of an estimate $\widehat{x}_n$ is measured by the difference of $f(\widehat{x}_n)$ and the *global minimum* of $f$:

$$\mathfrak{L}(\widehat{x}_n; f) := f(\widehat{x}_n) - f^* \qquad \text{where} \ \ f^* := \inf_{x \in \mathcal{X}} f(x). \tag{2}$$

Throughout the paper we take $\mathcal{X}$ to be the $d$-dimensional unit cube $[0,1]^d$, while our results can be easily generalized to other compact domains satisfying minimal regularity conditions.

When $f$ belongs to a smoothness class, say the Hölder class with exponent $\alpha$, a straightforward global optimization method is to first sample $n$ points uniformly at random from $\mathcal{X}$ and then construct

nonparametric estimates $\widehat{f}_n$ of $f$ using nonparametric regression methods such as (high-order) kernel smoothing or local polynomial regression [17, 46]. Classical analysis shows that the sup-norm reconstruction error $\|\widehat{f}_n - f\|_\infty = \sup_{x \in \mathcal{X}} |\widehat{f}_n(x) - f(x)|$ can be upper bounded by $\widetilde{O}_\mathbb{P}(n^{-\alpha/(2\alpha+d)})^2$. This global reconstruction guarantee then implies an $\widetilde{O}_\mathbb{P}(n^{-\alpha/(2\alpha+d)})$ upper bound on $\mathfrak{L}(\widehat{x}_n; f)$ by considering $\widehat{x}_n \in \mathcal{X}$ such that $\widehat{f}_n(\widehat{x}_n) = \inf_{x \in \mathcal{X}} \widehat{f}_n(x)$ (such an $\widehat{x}_n$ exists because $\mathcal{X}$ is closed and bounded). Formally, we have the following proposition (proved in the Appendix) that converts a global reconstruction guarantee into an upper bound on optimization error:

**Proposition 1.** *Suppose $\widehat{f}_n(\widehat{x}_n) = \inf_{x \in \mathcal{X}} \widehat{f}_n(x)$. Then $\mathfrak{L}(\widehat{x}_n; f) \leqslant 2\|\widehat{f}_n - f\|_\infty$.*

Typically, fundamental limits on the optimal optimization error are understood through the lens of *minimax analysis* where the object of study is the (global) minimax risk:

$$\inf_{\widehat{x}_n} \sup_{f \in \mathcal{F}} \mathbb{E}_f \mathfrak{L}(\widehat{x}_n, f), \tag{3}$$

where $\mathcal{F}$ is a certain smoothness function class such as the Hölder class. Although optimization appears to be easier than global reconstruction, we show that the $n^{-\alpha/(2\alpha+d)}$ rate is *not* improvable in the global minimax sense in Eq. (3) over Hölder classes. Such a surprising phenomenon was also noted in previous works [9, 22, 44] for related problems. On the other hand, extensive empirical evidence suggests that non-uniform/active allocations of query points can significantly reduce optimization error in practical global optimization of smooth, non-convex functions [40]. This raises the interesting question of understanding, from a theoretical perspective, under what conditions/in what scenarios is global optimization of smooth functions *easier* than their reconstruction, and the power of *active/feedback-driven* queries that play important roles in global optimization.

In this paper, we propose a theoretical framework that partially answers the above questions. In contrast to classical *global* minimax analysis of nonparametric estimation problems, we adopt a *local analysis* which characterizes the optimal convergence rate of optimization error when the underlying function $f$ is within the neighborhood of a "reference" function $f_0$. (See Sec. 2.2 for a rigorous formulation.) Our main results are to characterize the local convergence rates $R_n(f_0)$ for a wide range of reference functions $f_0 \in \mathcal{F}$. Our contributions can be summarized as follows:

1. We design an iterative (active) algorithm whose optimization error $\mathfrak{L}(\widehat{x}_n; f)$ converges at a rate of $R_n(f_0)$ depending on the reference function $f_0$. When the level sets of $f_0$ satisfy certain regularity and polynomial growth conditions, the local rate $R_n(f_0)$ can be upper bounded by $R_n(f_0) = \widetilde{O}(n^{-\alpha/(2\alpha+d-\alpha\beta)})$, where $\beta \in [0, d/\alpha]$ is a parameter depending on $f_0$ that characterizes the volume growth of *level sets* of $f_0$. (See assumption (A2), Proposition 2 and Theorem 1 for details). The rate matches the global minimax rate $n^{-\alpha/(2\alpha+d)}$ for worst-case $f_0$ where $\beta = 0$, but has the potential of being much faster when $\beta > 0$. We emphasize that our algorithm has no knowledge of $f_0$, $\alpha$ or $\beta$ and achieves this rate adaptively.

2. We prove *local* minimax lower bounds that match the $n^{-\alpha/(2\alpha+d-\alpha\beta)}$ upper bound, up to logarithmic factors in $n$. More specifically, we show that *even if $f_0$ is known*, no (active) algorithm can estimate $f$ in close neighborhoods of $f_0$ at a rate faster than $n^{-\alpha/(2\alpha+d-\alpha\beta)}$. We further show that, if active queries are not available and $x_1, \ldots, x_n$ are i.i.d. uniformly sampled from $\mathcal{X}$, the $n^{-\alpha/(2\alpha+d)}$ global minimax rate also applies locally regardless of how large $\beta$ is. Thus, there is an explicit gap between local minimax rates of active and uniform query models.

3. In the special case when $f$ is *convex*, the global optimization problem is usually referred to as *zeroth-order convex optimization* and this problem has been widely studied [1, 2, 6, 18, 24, 36]. Our results imply that, when $f_0$ is *strongly* convex and smooth, the local minimax rate $R_n(f_0)$ is on the order of $\widetilde{O}(n^{-1/2})$, which matches the convergence rates in [1]. Additionally, our negative results (Theorem 2) indicate that the $n^{-1/2}$ rate cannot be achieved if $f_0$ is merely convex, which seems to contradict $n^{-1/2}$ results in [2, 6] that do not require strong convexity of $f$. However, it should be noted that mere convexity of $f_0$ does *not* imply convexity of $f$ in a neighborhood of $f_0$ (e.g., $\|f - f_0\|_\infty \leqslant \varepsilon$). Our results show significant differences in the intrinsic difficulty of zeroth-order optimization of convex and near-convex functions.

## 1.1 Related Work

*Global optimization*, known variously as *black-box optimization*, *Bayesian optimization* and the *continuous-armed bandit*, has a long history in the optimization research community [25, 26] and has also received a significant amount of recent interest in statistics and machine learning [8, 9, 22, 31, 32, 40]. Many previous works [8, 28] have derived rates for non-convex smooth payoffs in "continuum-armed" bandit problems; however, they do not consider *local* rates specific to objective functions with certain growth conditions around the optima.

Among the existing works, [20, 34] is probably the closest to our paper, which studied a similar problem of estimating the set of all optima of a smooth function in Hausdorff's distance. For Hölder smooth functions with polynomial growth, [34] derives an $n^{-1/(2\alpha+d-\alpha\beta)}$ minimax rate for $\alpha < 1$ (later improved to $\alpha \geqslant 1$ in his thesis [33]), which is similar to our Propositions 2 and 3. [20, 34] also discussed adaptivity to unknown smoothness parameters. We however remark on several differences between our work and [34]. First, in [20, 34] only functions with polynomial growth are considered, while in our Theorems 1 and 2 functionals $\varepsilon_n^{\mathsf{U}}(f_0)$ and $\varepsilon_n^{\mathsf{L}}(f_0)$ are proposed for general reference functions $f_0$ satisfying mild regularity conditions, which include functions with polynomial growth as special cases. In addition, [34] considers the harder problem of estimating maxima sets in Hausdorff distance than producing a single approximate optima $\widehat{x}_T$. As a result, since the construction of minimax lower bound in [34] is no longer valid as an algorithm, without distinguishing between two functions with different optimal sets, can nevertheless produce a good approximate optimizer as long as the two functions under consideration have *overlapping* optimal sets. New constructions and information-theoretical techniques are therefore required to prove lower bounds under the weaker (one-point) approximate optimization framework. Finally, we prove a minimax lower bounds when only *uniform* query points are available and demonstrate a significant gap between algorithms having access to uniform or adaptively chosen data points.

[31, 32] impose additional assumptions on the level sets of the underlying function to obtain an improved convergence rate. The level set assumptions considered in the mentioned references are rather restrictive and essentially require the underlying function to be *uni-modal*, while our assumptions are much more flexible and apply to multi-modal functions as well. In addition, [31, 32] considered a *noiseless* setting in which exact function evaluations $f(x_t)$ can be obtained, while our paper studies the noise corrupted model in Eq. (1) for which vastly different convergence rates are derived. Finally, no matching lower bounds were proved in [31, 32].

[43] considered zeroth-order optimization of approximately convex functions and derived necessary and sufficient conditions for the convergence rates to be polynomial in domain dimension $d$.

The (stochastic) global optimization problem is similar to *mode estimation* of either densities or regression functions, which has a rich literature [13, 27, 39]. An important difference between statistical mode estimation and global optimization is the way sample/query points $x_1, \ldots, x_n \in \mathcal{X}$ are distributed: in mode estimation it is customary to assume the samples are independently and identically distributed, while in global optimization sequential designs of samples/queries are allowed. Furthermore, to estimate/locate the mode of an unknown density or regression function, such a mode has to be well-defined; on the other hand, producing an estimate $\widehat{x}_n$ with small $\mathfrak{L}(\widehat{x}_n, f)$ is easier and results in weaker conditions imposed on the underlying function.

Methodology-wise, our iterative procedure also resembles disagreement-based active learning methods [5, 14, 21]. The intermediate steps of candidate point elimination can also be viewed as sequences of level set estimation problems [38, 42, 45] or cluster tree estimation [4, 12] with active queries.

Another line of research has focused on *first-order* optimization of quasi-convex or non-convex functions [3, 10, 19, 23, 37, 48], in which exact or unbiased evaluations of function *gradients* are available at query points $x \in \mathcal{X}$. [48] considered a Cheeger's constant restriction on level sets which is similar to our level set regularity assumptions (A2 and A2'). [15, 16] studied local minimax rates of first-order optimization of convex functions. First-order optimization differs significantly from our setting because unbiased gradient estimation is generally impossible in the model of Eq. (1). Furthermore, most works on (first-order) non-convex optimization focus on convergence to stationary points or local minima, while we consider convergence to global minima.

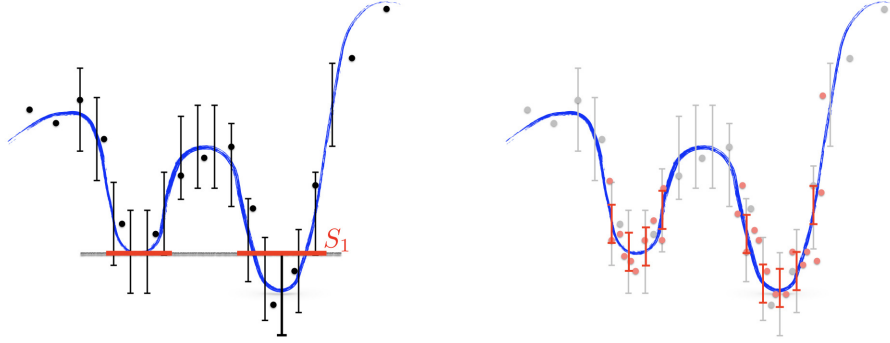

Figure 1: Informal illustrations of our algorithm that attains Theorem 1 (details in the appendix). Solid blue curves depict the underlying function $f$ to be optimized, black and red solid dots denote the query points and their responses $\{(x_t, y_t)\}$, and black/red vertical line segments correspond to uniform confidence intervals on function evaluations constructed using current batch of data observed. The left figure illustrates the first epoch of our algorithm, where query points are uniformly sampled from the entire domain $\mathcal{X}$. Afterwards, sub-optimal locations based on constructed confidence intervals are removed, and a shrinkt "candidate set" $S_1$ is obtained. The algorithm then proceeds to the second epoch, illustrated in the right figure, where query points (in red) are sampled only from the restricted candidate set and shorter confidence intervals (also in red) are constructed and updated. The procedure is repeated until $O(\log n)$ epochs are completed.

## 2  Background and Notation

We first review standard asymptotic notation that will be used throughout this paper. For two sequences $\{a_n\}_{n=1}^{\infty}$ and $\{b_n\}_{n=1}^{\infty}$, we write $a_n = O(b_n)$ or $a_n \lesssim b_n$ if $\limsup_{n\to\infty} |a_n|/|b_n| < \infty$, or equivalently $b_n = \Omega(a_n)$ or $b_n \gtrsim a_n$. Denote $a_n = \Theta(b_n)$ or $a_n \asymp b_n$ if both $a_n \lesssim b_n$ and $a_n \gtrsim b_n$ hold. We also write $a_n = o(b_n)$ or equivalently $b_n = \omega(a_n)$ if $\lim_{n\to\infty} |a_n|/|b_n| = 0$. For two sequences of random variables $\{A_n\}_{n=1}^{\infty}$ and $\{B_n\}_{n=1}^{\infty}$, denote $A_n = O_{\mathbb{P}}(B_n)$ if for every $\epsilon > 0$, there exists $C > 0$ such that $\limsup_{n\to\infty} \Pr[|A_n| > C|B_n|] \leqslant \epsilon$. For $r > 0$, $1 \leqslant p \leqslant \infty$ and $x \in \mathbb{R}^d$, we denote $B_r^p(x) := \{z \in \mathbb{R}^d : \|z - x\|_p \leqslant r\}$ as the $d$-dimensional $\ell_p$-ball of radius $r$ centered at $x$, where the vector $\ell_p$ norm is defined as $\|x\|_p := (\sum_{j=1}^d |x_j|^p)^{1/p}$ for $1 \leqslant p < \infty$ and $\|x\|_{\infty} := \max_{1 \leqslant j \leqslant d} |x_j|$. For any subset $S \subseteq \mathbb{R}^d$ we denote by $B_r^p(x; S)$ the set $B_r^p(x) \cap S$.

### 2.1  Passive and Active Query Models

Let $U$ be a known random quantity defined on a probability space $\mathcal{U}$. The following definitions characterize all passive and active optimization algorithms:

**Definition 1** (The passive query model)**.** *Let $x_1, \ldots, x_n$ be i.i.d. points uniformly sampled on $\mathcal{X}$ and $y_1, \ldots, y_n$ be observations from the model Eq. (1). A passive optimization algorithm $\mathcal{A}$ with $n$ queries is parameterized by a mapping $\phi_n : (x_1, y_1, \ldots, x_n, y_n, U) \mapsto \widehat{x}_n$ that maps the i.i.d. observations $\{(x_i, y_i)\}_{i=1}^n$ to an estimated optimum $\widehat{x}_n \in \mathcal{X}$, potentially randomized by $U$.*

**Definition 2** (The active query model)**.** *An active optimization algorithm can be parameterized by mappings $(\chi_1, \ldots, \chi_n, \phi_n)$, where for $t = 1, \ldots, n$,*

$$\chi_t : (x_1, y_1, \ldots, x_{t-1}, y_{t-1}, U) \mapsto x_t$$

*produces a query point $x_t \in \mathcal{X}$ based on previous observations $\{(x_i, t_i)\}_{i=1}^{t-1}$, and*

$$\phi_n : (x_1, y_1, \ldots, x_n, y_n, U) \mapsto \widehat{x}_n$$

*produces the final estimate. All mappings $(\chi_1, \ldots, \chi_n, \phi_n)$ can be randomized by $U$.*

### 2.2  Local Minimax Rates

We use the classical *local minimax analysis* [47] to understand the fundamental information-theoretical limits of noisy global optimization of smooth functions. On the upper bound side,

we seek (active) estimators $\widehat{x}_n$ such that

$$\sup_{f_0 \in \Theta} \sup_{f \in \Theta', \|f - f_0\|_\infty \leqslant \varepsilon_n(f_0)} \Pr_f \left[ \mathfrak{L}(\widehat{x}_n; f) \geqslant C_1 \cdot R_n(f_0) \right] \leqslant 1/4, \tag{4}$$

where $C_1 > 0$ is a positive constant. Here $f_0 \in \Theta$ is referred to as the *reference function*, and $f \in \Theta'$ is the true underlying function which is assumed to be "near" $f_0$. The minimax convergence rate of $\mathfrak{L}(\widehat{x}_n; f)$ is then characterized *locally* by $R_n(f_0)$ which depends on the reference function $f_0$. The constant of $1/4$ is chosen arbitrarily and any small constant leads to similar conclusions. To establish negative results (i.e., locally minimax lower bounds), in contrast to the upper bound formulation, we assume the potential active optimization estimator $\widehat{x}_n$ has *perfect knowledge* about the reference function $f_0 \in \Theta$. We then prove locally minimax lower bounds of the form

$$\inf_{\widehat{x}_n} \sup_{f \in \Theta', \|f - f_0\|_\infty \leqslant \varepsilon_n(f_0)} \Pr_f \left[ \mathfrak{L}(\widehat{x}_n; f) \geqslant C_2 \cdot R_n(f_0) \right] \geqslant 1/3, \tag{5}$$

where $C_2 > 0$ is another positive constant and $\varepsilon_n(f_0), R_n(f_0)$ are desired local convergence rates for functions near the reference $f_0$.

Although in some sense classical, the local minimax definition we propose warrants further discussion.

1. **Roles of $\Theta$ and $\Theta'$:** The reference function $f_0$ and the true functions $f$ are assumed to belong to different but closely related function classes $\Theta$ and $\Theta'$. In particular, in our paper $\Theta \subseteq \Theta'$, meaning that less restrictive assumptions are imposed on the true underlying function $f$ compared to those imposed on the reference function $f_0$ on which $R_n$ and $\varepsilon_n$ are based.

2. **Upper Bounds:** It is worth emphasizing that the estimator $\widehat{x}_n$ has no knowledge of the reference function $f_0$. From the perspective of upper bounds, we can consider the simpler task of producing $f_0$-dependent bounds (eliminating the second supremum) to instead study the (already interesting) quantity:

$$\sup_{f_0 \in \Theta} \Pr_{f_0} \left[ \mathfrak{L}(\widehat{x}_n; f_0) \geqslant C_1 R_n(f_0) \right] \leqslant 1/4.$$

   As indicated above we maintain the double-supremum in the definition because fewer assumptions are imposed directly on the true underlying function $f$, and further because it allows to more directly compare our upper and lower bounds.

3. **Lower Bounds and the choice of the "localization radius" $\varepsilon_n(f_0)$:** Our lower bounds allow the estimator knowledge of the reference function (this makes establishing the lower bound more challenging). Eq. (5) implies that no estimator $\widehat{x}_n$ can effectively optimize a function $f$ close to $f_0$ beyond the convergence rate of $R_n(f_0)$, even if perfect knowledge of the reference function $f_0$ is available a priori. The $\varepsilon_n(f_0)$ parameter that decides the "range" in which local minimax rates apply is taken to be on the same order as the actual local rate $R_n(f_0)$ in this paper. This is (up to constants) the smallest radius for which we can hope to obtain non-trivial lower-bounds: if we consider a much smaller radius than $R_n(f_0)$ then the trivial estimator which outputs the minimizer of the reference function would achieve a faster rate than $R_n(f_0)$. Selecting the smallest possible radius makes establishing the lower bound most challenging but provides a refined picture of the complexity of zeroth-order optimization.

## 3 Main Results

With this background in place we now turn our attention to our main results. We begin by collecting our assumptions about the true underlying function and the reference function in Section 3.1. We state and discuss the consequences of our upper and lower bounds in Sections 3.2 and 3.3 respectively. We defer most technical proofs to the Appendix and turn our attention to our optimization algorithm in Section A.

### 3.1 Assumptions

We first state and motivate assumptions that will be used. The first assumption states that $f$ is locally Hölder smooth on its level sets.

(A1) There exist constants $\kappa, \alpha, M > 0$ such that $f$ restricted on $\mathcal{X}_{f,\kappa} := \{x \in \mathcal{X} : f(x) \leqslant f^* + \kappa\}$ belongs to the Hölder class $\Sigma^\alpha(M)$, meaning that $f$ is $k$-times differentiable on $\mathcal{X}_{f,\kappa}$ and furthermore for any $x, x' \in \mathcal{X}_{f,\kappa}$, [3]

$$\sum_{j=0}^{k} \sum_{\alpha_1+\ldots+\alpha_d=j} |f^{(\boldsymbol{\alpha},j)}(x)| + \sum_{\alpha_1+\ldots+\alpha_d=k} \frac{|f^{(\boldsymbol{\alpha},k)}(x) - f^{(\boldsymbol{\alpha},k)}(x')|}{\|x-x'\|_\infty^{\alpha-k}} \leqslant M. \quad (6)$$

Here $k = \lfloor \alpha \rfloor$ is the largest integer lower bounding $\alpha$ and $f^{(\boldsymbol{\alpha},j)}(x) := \partial^j f(x)/\partial x_1^{\alpha_1} \ldots \partial x_d^{\alpha_d}$.

We use $\Sigma_\kappa^\alpha(M)$ to denote the class of all functions satisfying (A1). We remark that (A1) is weaker than the standard assumption that $f$ on its entire domain $\mathcal{X}$ belongs to the Hölder class $\Sigma^\alpha(M)$. This is because places with function values larger than $f^* + \kappa$ can be easily detected and removed by a pre-processing step. We give further details of the pre-processing step in Section A.3.

Our next assumption concern the "regularity" of the *level sets* of the "reference" function $f_0$. Define $L_{f_0}(\epsilon) := \{x \in \mathcal{X} : f_0(x) \leqslant f_0^* + \epsilon\}$ as the $\epsilon$-level set of $f_0$, and $\mu_{f_0}(\epsilon) := \lambda(L_{f_0}(\epsilon))$ as the Lebesgue measure of $L_{f_0}(\epsilon)$, also known as the *distribution function*. Define also $N(L_{f_0}(\epsilon), \delta)$ as the smallest number of $\ell_2$-balls of radius $\delta$ that cover $L_{f_0}(\epsilon)$.

(A2) There exist constants $c_0 > 0$ and $C_0 > 0$ such that $N(L_{f_0}(\epsilon), \delta) \leqslant C_0[1 + \mu_{f_0}(\epsilon)\delta^{-d}]$ for all $\epsilon, \delta \in (0, c_0]$.

We use $\Theta_{\mathbf{C}}$ to denote all functions that satisfy (A2) with respect to parameters $\mathbf{C} = (c_0, C_0)$.

At a higher level, the regularity condition (A2) assumes that the level sets are sufficiently "regular" such that covering them with small-radius balls does not require significantly larger total volumes. For example, consider a perfectly regular case of $L_{f_0}(\epsilon)$ being the $d$-dimensional $\ell_2$ ball of radius $r$: $L_{f_0}(\epsilon) = \{x \in \mathcal{X} : \|x - x^*\|_2 \leqslant r\}$. Clearly, $\mu_{f_0}(\epsilon) \asymp r^d$. In addition, the $\delta$-covering number in $\ell_2$ of $L_{f_0}(\epsilon)$ is on the order of $1 + (r/\delta)^d \asymp 1 + \mu_{f_0}(\epsilon)\delta^{-d}$, which satisfies the scaling in (A2).

When (A2) holds, uniform confidence intervals of $f$ on its level sets are easy to construct because little statistical efficiency is lost by slightly enlarging the level sets so that complete $d$-dimensional cubes are contained in the enlarged level sets. On the other hand, when regularity of level sets fails to hold such nonparametric estimation can be very difficult or even impossible. As an extreme example, suppose the level set $L_{f_0}(\epsilon)$ consists of $\mathfrak{n}$ standalone and well-spaced points in $\mathcal{X}$: the Lebesgue measure of $L_{f_0}(\epsilon)$ would be zero, but at least $\Omega(\mathfrak{n})$ queries are necessary to construct uniform confidence intervals on $L_{f_0}(\epsilon)$. It is clear that such $L_{f_0}(\epsilon)$ violates (A2), because $N(L_{f_0}(\epsilon), \delta) \geqslant \mathfrak{n}$ as $\delta \to 0^+$ but $\mu_{f_0}(\epsilon) = 0$.

## 3.2 Upper Bound

The following theorem is our main result that upper bounds the local minimax rate of noisy global optimization with active queries.

**Theorem 1.** *For any $\alpha, M, \kappa, c_0, C_0 > 0$ and $f_0 \in \Sigma_\kappa^\alpha(M) \cap \Theta_{\mathbf{C}}$, where $\mathbf{C} = (c_0, C_0)$, define*

$$\varepsilon_n^{\mathsf{U}}(f_0) := \sup\left\{\varepsilon > 0 : \varepsilon^{-(2+d/\alpha)}\mu_{f_0}(\varepsilon) \geqslant n/\log^\omega n\right\}, \quad (7)$$

*where $\omega > 5 + d/\alpha$ is a large constant. Suppose also that $\varepsilon_n^{\mathsf{U}}(f_0) \to 0$ as $n \to \infty$. Then for sufficiently large $n$, there exists an estimator $\widehat{x}_n$ with access to $n$ active queries $x_1, \ldots, x_n \in \mathcal{X}$, a constant $C_R > 0$ depending only on $\alpha, M, \kappa, c, c_0, C_0$ and a constant $\gamma > 0$ depending only on $\alpha$ and $d$ such that*

$$\sup_{f_0 \in \Sigma_\kappa^\alpha(M) \cap \Theta_{\mathbf{C}}} \sup_{f \in \Sigma_\kappa^\alpha(M), \|f-f_0\|_\infty \leqslant \varepsilon_n^{\mathsf{U}}(f_0)} \Pr_f\left[\mathcal{L}(\widehat{x}_n, f) > C_R \log^\gamma n \cdot (\varepsilon_n^{\mathsf{U}}(f_0) + n^{-1/2})\right] \leqslant 1/4.$$
$$(8)$$

*Remark* 1. Unlike the (local) smoothness class $\Sigma_\kappa^\alpha(M)$, the additional function class $\Theta_{\mathbf{C}}$ that encapsulates (A2) is imposed only on the "reference" function $f_0$ but not the true function $f$ to be estimated. This makes the assumptions considerably weaker because the true function $f$ may violate (A2) while our results remain valid.

*Remark* 2. The estimator $\widehat{x}_n$ does *not* require knowledge of parameters $\kappa, c_0, C_0$ or $\varepsilon_n^{\mathsf{U}}(f_0)$, and automatically adapts to them, as shown in the next section. While the knowledge of smoothness parameters $\alpha$ and $M$ seems to be necessary, we remark that it is possible to adapt to $\alpha$ and $M$ by running $O(\log^2 n)$ parallel sessions of $\widehat{x}_n$ on $O(\log n)$ grids of $\alpha$ and $M$ values, and then using $\Omega(n/\log^2 n)$ single-point queries to decide on the location with the smallest function value. Such an adaptive strategy was suggested in [20] to remove an additional condition in [34], which also applies to our settings.

*Remark* 3. By repeating the algorithm independently for $t$ times and using the "multiple query" strategy in the above remark, the failure probability of our proposed algorithm can be reduced to as small as $4^{-t}$, an *exponentially* decaying probability with respect to repetitions $t$.

*Remark* 4. When the distribution function $\mu_{f_0}(\epsilon)$ does not change abruptly with $\epsilon$ the expression of $\varepsilon_n^{\mathsf{U}}(f_0)$ can be significantly simplified. In particular, if for all $\epsilon \in (0, c_0]$ it holds that

$$\mu_{f_0}(\epsilon/\log n) \geq \mu_{f_0}(\epsilon)/[\log n]^{O(1)}, \tag{9}$$

then $\varepsilon_n^{\mathsf{U}}(f_0)$ can be upper bounded as

$$\varepsilon_n^{\mathsf{U}}(f_0) \leq [\log n]^{O(1)} \cdot \sup\left\{\varepsilon > 0 : \varepsilon^{-(2+d/\alpha)}\mu_{f_0}(\varepsilon) \geq n\right\}. \tag{10}$$

It is also noted that if $\mu_{f_0}(\epsilon)$ has a polynomial behavior of $\mu_{f_0}(\epsilon) \asymp \epsilon^\beta$ for some constant $\beta \geq 0$, then Eq. (9) is satisfied and so is Eq. (10).

The quantity $\varepsilon_n^{\mathsf{U}}(f_0) = \inf\{\varepsilon > 0 : \varepsilon^{-(2+d/\alpha)}\mu_{f_0}(\varepsilon) \geq n/\log^\omega n\}$ is crucial in determining the convergence rate of optimization error of $\widehat{x}_n$ *locally* around the reference function $f_0$. While the definition of $\varepsilon_n^{\mathsf{U}}(f_0)$ is mostly implicit and involves solving an inequality concerning the distribution function $\mu_{f_0}(\cdot)$, we remark that it admits a simple form when $\mu_{f_0}$ has a polynomial growth rate similar to a local Tsybakov noise condition [29, 46], as shown by the following proposition:

**Proposition 2.** *Suppose* $\mu_{f_0}(\epsilon) \lesssim \epsilon^\beta$ *for some constant* $\beta \in [0, 2 + d/\alpha)$. *Then* $\varepsilon_n^{\mathsf{U}}(f_0) = \widetilde{O}(n^{-\alpha/(2\alpha+d-\alpha\beta)})$. *In addition, if* $\beta \in [0, d/\alpha]$ *then* $\varepsilon_n^{\mathsf{U}}(f_0) + n^{-1/2} \lesssim \varepsilon_n^{\mathsf{U}}(f_0) = \widetilde{O}(n^{-\alpha/(2\alpha+d-\alpha\beta)})$.

We remark that the condition $\beta \in [0, d/\alpha]$ was also adopted in the previous work [34, Remark 6] Also, for Lipschitz continuous functions ($\alpha = 1$) our conditions are similar to [20] and implies a corresponding near-optimality dimension $d'$ considered in [20].

Proposition 2 can be easily verified by solving the system $\varepsilon^{-(2+d/\alpha)}\mu_{f_0}(\varepsilon) \geq n/\log^\omega n$ with the condition $\mu_{f_0}(\epsilon) \lesssim \epsilon^\beta$. We therefore omit its proof. The following two examples give some simple reference functions $f_0$ that satisfy the $\mu_{f_0}(\epsilon) \lesssim \epsilon^\beta$ condition in Proposition 2 with particular values of $\beta$.

*Example* 1. The constant function $f_0 \equiv 0$ satisfies (A1), (A2) and the condition in Proposition 2 with $\beta = 0$.

*Example* 2. $f_0 \in \Sigma_\kappa^2(M)$ that is *strongly convex* [4] satisfies (A1), (A2) and the condition in Proposition 2 with $\beta = d/2$.

Example 1 is simple to verify, as the volume of level sets of the constant function $f_0 \equiv 0$ exhibits a phase transition at $\epsilon = 0$ and $\epsilon > 0$, rendering $\beta = 0$ the only parameter option for which $\mu_{f_0}(\epsilon) \lesssim \epsilon^\beta$. Example 2 is more involved, and holds because the strong convexity of $f_0$ *lower bounds* the growth rate of $f_0$ when moving away from its minimum. We give a rigorous proof of Example 2 in the appendix. We also remark that $f_0$ does *not* need to be exactly strongly convex for $\beta = d/2$ to hold, and the example is valid for, e.g., piecewise strongly convex functions with a constant number of pieces too.

To best interpret the results in Theorem 1 and Proposition 2, it is instructive to compare the "local" rate $n^{-\alpha/(2\alpha+d-\alpha\beta)}$ with the baseline rate $n^{-\alpha/(2\alpha+d)}$, which can be attained by reconstructing $f$

in sup-norm and applying Proposition 1. Since $\beta \geqslant 0$, the local convergence rate established in Theorem 1 is never slower, and the improvement compared to the baseline rate $n^{-\alpha/(2\alpha+d)}$ is dictated by $\beta$, which governs the growth rate of volume of level sets of the reference function $f_0$. In particular, for functions that grows fast when moving away from its minimum, the parameter $\beta$ is large and therefore the local convergence rate around $f_0$ could be much faster than $n^{-\alpha/(2\alpha+d)}$.

Theorem 1 also implies concrete convergence rates for special functions considered in Examples 1 and 2. For the constant reference function $f_0 \equiv 0$, Example 1 and Theorem 1 yield that $R_n(f_0) \asymp n^{-\alpha/(2\alpha+d)}$, which matches the baseline rate $n^{-\alpha/(2\alpha+d)}$ and suggests that $f_0 \equiv 0$ is the worst-case reference function. This is intuitive, because $f_0 \equiv 0$ has the most drastic level set change at $\epsilon \to 0^+$ and therefore small perturbations anywhere of $f_0$ result in changes of the optimal locations. On the other hand, if $f_0$ is strongly smooth and convex as in Example 2, Theorem 1 suggests that $R_n(f_0) \asymp n^{-1/2}$, which is significantly better than the $n^{-2/(4+d)}$ baseline rate [5] and also matches existing works on zeroth-order optimization of convex functions [1]. The faster rate holds intuitively because strongly convex functions grows fast when moving away from the minimum, which implies small level set changes. An active query algorithm could then focus most of its queries onto the small level sets of the underlying function, resulting in more accurate local function reconstructions and faster optimization error rate.

Our proof of Theorem 1 is constructive, by upper bounding the local minimax optimization error of an explicit algorithm. At a higher level, the algorithm partitions the $n$ active queries evenly into $\log n$ epochs, and level sets of $f$ are estimated at the end of each epoch by comparing (uniform) confidence intervals on a dense grid on $\mathcal{X}$. It is then proved that the volume of the estimated level sets contracts *geometrically*, until the target convergence rate $R_n(f_0)$ is attained.

### 3.3 Lower Bounds

We prove local minimax lower bounds that match the upper bounds in Theorem 1 up to logarithmic terms. As we remarked in Section 2.2, in the local minimax lower bound formulation we assume the data analyst has full knowledge of the reference function $f_0$, which makes the lower bounds stronger as more information is available a priori.

To facilitate such a strong local minimax lower bounds, the following additional condition is imposed on the reference function $f_0$ of which the data analyst has perfect information.

(A2') There exist constants $c_0', C_0' > 0$ such that $M(L_{f_0}(\epsilon), \delta) \geqslant C_0' \mu_{f_0}(\epsilon)\delta^{-d}$ for all $\epsilon, \delta \in (0, c_0']$, where $M(L_{f_0}(\epsilon), \delta)$ is the maximum number of disjoint $\ell_2$ balls of radius $\delta$ that can be packed into $L_{f_0}(\epsilon)$.

We denote $\Theta_{\mathbf{C}'}'$ as the class of functions that satisfy (A2') with respect to parameters $\mathbf{C}' = (c_0', C_0') > 0$. Intuitively, (A2') can be regarded as the "reverse" version of (A2), which basically means that (A2) is "tight".

We are now ready to state our main negative result, which shows, from an information-theoretical perspective, that the upper bound in Theorem 1 is not improvable.

**Theorem 2.** *Suppose $\alpha, c_0, C_0, c_0', C_0' > 0$ and $\kappa = \infty$. Denote $\mathbf{C} = (c_0, C_0)$ and $\mathbf{C}' = (c_0', C_0')$. For any $f_0 \in \Theta_{\mathbf{C}} \cap \Theta_{\mathbf{C}'}'$, define*

$$\varepsilon_n^{\mathsf{L}}(f_0) := \sup\left\{\varepsilon > 0 : \varepsilon^{-(2+d/\alpha)}\mu_{f_0}(\varepsilon) \geqslant n\right\}. \tag{11}$$

*Then there exist constant $M > 0$ depending on $\alpha, d, \mathbf{C}, \mathbf{C}'$ such that, for any $f_0 \in \Sigma_\kappa^\alpha(M/2) \cap \Theta_{\mathbf{C}} \cap \Theta_{\mathbf{C}'}'$,*

$$\inf_{\widehat{x}_n} \sup_{f \in \Sigma_\kappa^\alpha(M), \|f-f_0\|_\infty \leqslant 2\varepsilon_n^{\mathsf{L}}(f_0)} \Pr_f\left[\mathfrak{L}(\widehat{x}_n; f) \geqslant \varepsilon_n^{\mathsf{L}}(f_0)\right] \geqslant \frac{1}{3}. \tag{12}$$

*Remark* 5. For any $f_0$ and $n$ it always holds that $\varepsilon_n^{\mathsf{L}}(f_0) \leqslant \varepsilon_n^{\mathsf{U}}(f_0)$.

*Remark* 6. If the distribution function $\mu_{f_0}(\epsilon)$ satisfies Eq. (9) in Remark 4, then $\varepsilon_n^{\mathsf{L}}(f_0) \geqslant \varepsilon_n^{\mathsf{U}}(f_0)/[\log n]^{O(1)}$.

*Remark* 7. As the upper bound in Theorem 1 might depends *exponentially* on domain dimension $d$, there might also be an exponential gap of $d$ between the upper and lower bounds established in Theorems 1 and 2.

Remark 5 shows that there might be a gap between the locally minimax upper and lower bounds in Theorems 1 and 2. Nevertheless, Remark 6 shows that under the mild condition of $\mu_{f_0}(\epsilon)$ does not change too abruptly with $\epsilon$, the gap between $\varepsilon_n^{\mathsf{U}}(f_0)$ and $\varepsilon_n^{\mathsf{L}}(f_0)$ is only a poly-logarithmic term in $n$. Additionally, the following proposition derives explicit expression of $\varepsilon_n^{\mathsf{L}}(f_0)$ for reference functions whose distribution functions have a polynomial growth, which matches the Proposition 2 up to $\log n$ factors. Its proof is again straightforward.

**Proposition 3.** *Suppose $\mu_{f_0}(\epsilon) \gtrsim \epsilon^\beta$ for some $\beta \in [0, 2 + d/\alpha)$. Then $\varepsilon_n^{\mathsf{L}}(f_0) = \Omega(n^{-\alpha/(2\alpha+d-\alpha\beta)})$.*

The following proposition additionally shows the existence of $f_0 \in \Sigma_\infty^\alpha(M) \cap \Theta_{\mathbf{C}} \cap \Theta_{\mathbf{C}'}$ that satisfies $\mu_{f_0}(\epsilon) \asymp \epsilon^\beta$ for any values of $\alpha > 0$ and $\beta \in [0, d/\alpha]$. Its proof is given in the appendix.

**Proposition 4.** *Fix arbitrary $\alpha, M > 0$ and $\beta \in [0, d/\alpha]$. There exists $f_0 \in \Sigma_\kappa^\alpha(M) \cap \Theta_{\mathbf{C}} \cap \Theta_{\mathbf{C}'}$ for $\kappa = \infty$ and constants $\mathbf{C} = (c_0, C_0)$, $\mathbf{C}' = (c_0', C_0')$ that depend only on $\alpha, \beta, M$ and $d$ such that $\mu_{f_0}(\epsilon) \asymp \epsilon^\beta$.*

Theorem 2 and Proposition 3 show that the $n^{-\alpha/(2\alpha+d-\alpha\beta)}$ upper bound on local minimax convergence rate established in Theorem 1 is not improvable up to logarithmic factors of $n$. Such information-theoretical lower bounds on the convergence rates hold *even if the data analyst has perfect information of $f_0$*, the reference function on which the $n^{-\alpha/(2\alpha+d-\alpha\beta)}$ local rate is based. Our results also imply an $n^{-\alpha/(2\alpha+d)}$ minimax lower bound over all $\alpha$-Hölder smooth functions, showing that without additional assumptions, noisy optimization of smooth functions is as difficult as reconstructing the unknown function in sup-norm.

Our proof of Theorem 2 also differs from existing minimax lower bound proofs for active nonparametric models [11]. The classical approach is to invoke Fano's inequality and to upper bound the KL divergence between different underlying functions $f$ and $g$ using $\|f - g\|_\infty$, corresponding to the point $x \in \mathcal{X}$ that leads to the largest KL divergence. Such an approach, however, does not produce tight lower bounds for our problem. To overcome such difficulties, we borrow the lower bound analysis for bandit pure exploration problems in [7]. In particular, our analysis considers the query distribution of any active query algorithm $\mathcal{A} = (\varphi_1, \ldots, \varphi_n, \phi_n)$ under the reference function $f_0$ and bounds the perturbation in query distributions between $f_0$ and $f$ using Le Cam's lemma. Afterwards, an adversarial function choice $f$ can be made based on the query distributions of the considered algorithm $\mathcal{A}$.

Theorem 2 applies to any global optimization method that makes *active* queries, corresponding to the query model in Definition 2. The following theorem, on the other hand, shows that for passive algorithms (Definition 1) the $n^{-\alpha/(2\alpha+d)}$ optimization rate is not improvable even with additional level set assumptions imposed on $f_0$. This demonstrates an explicit gap between passive and adaptive query models in global optimization problems.

**Theorem 3.** *Suppose $\alpha, c_0, C_0, c_0', C_0' > 0$ and $\kappa = \infty$. Denote $\mathbf{C} = (c_0, C_0)$ and $\mathbf{C}' = (c_0', C_0')$. Then there exist constant $M > 0$ depending on $\alpha, d, \mathbf{C}, \mathbf{C}'$ and $N$ depending on $M$ such that, for any $f_0 \in \Sigma_\kappa^\alpha(M/2) \cap \Theta_{\mathbf{C}} \cap \Theta_{\mathbf{C}'}$ satisfying $\varepsilon_n^{\mathsf{L}}(f_0) \leqslant \tilde{\varepsilon}_n^{\mathsf{L}} =: [\log n / n]^{\alpha/(2\alpha+d)}$,*

$$\inf_{\breve{x}_n} \sup_{f \in \Sigma_\kappa^\alpha(M), \|f - f_0\|_\infty \leqslant 2\tilde{\varepsilon}_n^{\mathsf{L}}} \Pr_f \left[ \mathfrak{L}(\hat{x}_n; f) \geqslant \tilde{\varepsilon}_n^{\mathsf{L}} \right] \geqslant \frac{1}{3} \quad \text{for all } n \geqslant N. \tag{13}$$

Intuitively, the apparent gap demonstrated by Theorems 2 and 3 between the active and passive query models stems from the observation that, a passive algorithm $\mathcal{A}$ only has access to uniformly sampled query points $x_1, \ldots, x_n$ and therefore cannot focus on a small level set of $f$ in order to improve query efficiency. In addition, for functions that grow faster when moving away from their minima (implying a larger value of $\beta$), the gap between passive and active query models becomes bigger as active queries can more effectively exploit the restricted level sets of such functions.

## 4 Conclusion

In this paper we consider the problem of noisy zeroth-order optimization of general smooth functions. Matching lower and upper bounds on the local minimax convergence rates are established, which are significantly different from classical minimax rates in nonparametric regression problems. Many interesting future directions exist along this line of research, including exploitation of additive structures in the underlying function $f$ to completely remove curse of dimensionality, functions with spatially heterogeneous smoothness or level set growth behaviors, and to design more computationally efficient algorithms that work well in practice.

## Acknowledgement

This work is supported by AFRL grant FA8750-17-2-0212. We thank the anonymous reviewers for many helpful suggestions that improved the presentation of this paper.

## Footnotes

[1]The exact distribution of $\varepsilon_t$ is not important, and our results hold for sub-Gaussian noise too.

[2]In the $\widetilde{O}(\cdot)$ or $\widetilde{O}_\mathbb{P}(\cdot)$ notation we drop poly-logarithmic dependency on $n$

[3] the particular $\ell_\infty$ norm is used for convenience only and can be replaced by any equivalent vector norms.

[4]A twice differentiable function $f_0$ is strongly convex if $\exists \sigma > 0$ such that $\nabla^2 f_0(x) \geq \sigma I, \forall x \in \mathcal{X}$.

[5] Note that $f_0$ being strongly smooth implies $\alpha = 2$ in the local smoothness assumption.

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
