[Reviews · NeurIPS 2018]

Reviewer 1



This paper studies local minimax bounds for noisy optimization of smooth functions. In the traditional setting of global optimization, the algorithmic goal is to design an adaptive procedure to find an approximate (in this case, global) optimum of an unknown function f, given access to noisy evaluations at feasible points. This notion of complexity might be too rough to understand the difficulty of the problem, so it is proposed to study the local minimax complexity, when the algorithm may have additional information on how close is the objective to a fixed function f_0. In practice, the algorithm doesn't need to know f_0, but this parameterization serves as an instance-dependent notion of complexity. This paper additionally considers how the regularity of the function (given by its Holder continuity) improves the complexity, as it is well-known that for Lipschitz functions adaptivity does not help. The main contributions of the paper can be summarized as follows: 1. An adaptive algorithm whose optimization error decreases at a rate R_n(f_0)=\tildeO(n^{-\alpha/[2\alpha+d-\alpha\beta]}), where \alpha is the Holder exponent and beta is a parameter related to the volume growth of level sets of f_0. This algorithm does not need to know f_0 in advance. 2. A local minimax lower bound matching the previous upper bound up to polylogs, for the case when even f_0 is known in advance. 3. A lower bound of Omega(n^{-\alpha/[2\alpha+d]}) for nonadaptive algorithms, exposing a substantial gap between adaptive and nonadaptive methods. 4. For the case when f_0 is strongly convex they show a local minimax rate of \tilde O(1/\sqrt{n}), and a stricly stronger lower bound when f_0 is merely convex. The paper is quite interesting, and looks mathematically solid. I didn't have the chance to carefully verify all proofs, but they seem correct. Detailed comments: 1. The setup of the paper seems comparable to the one of approximately convex optimization (see e.g. https://papers.nips.cc/paper/5841-information-theoretic-lower-bounds-for-convex-optimization-with-erroneous-oracles or https://papers.nips.cc/paper/6576-algorithms-and-matching-lower-bounds-for-approximately-convex-optimization.pdf). This setting is analog to the one proposed in this paper, but without the extra noise in function evaluations. In particular, they show a phase transition of epsilon_n(f_0) (for some f_0) where no polynomial number of adaptive queries can reach an approximate solution. Can the results of the paper be translated to this zero noise setting? -------------- After Rebuttal -------------- I want to thank the authors for clarifying the differences w.r.t. approximate convex optimization. It is very interesting how these subtleties change complexity bounds. As some other reviewer suggested, it is important that the authors also include the dimension dependence in the minimax rates.

Reviewer 2



This paper is a theoretical study of global stochastic optimization of an unknown non-convex function under smoothness conditions in a local minimax setting under active and passive sampling. The paper is presented in the setting where the noise is (sub-)Gaussian. The references and review of prior work is adequate in the context of black box optimization (particularly in conjunction with the appendix), and is of interest to the NIPS theory community with interests in non-convex optimization (as can be seen in the reference list) and active learning as well as the theoretical statistics community. The main original contribution is a precise analysis of the local minimax setting (by introducing the appropriate framework), and showing that the local minimax rates of convergence can be significantly better than the global minimax setting under a Holder-type smoothness and regularity condition (A1,A2) in Section 3.1. The paper does a good job of presenting some intuition for the results particularly in the context of comparison to the analogous problem for (strongly) convex optimization. The paper is overall well written, with minor typos and formatting issues. The paper could do a better job at pointing to the appropriate appendices in the body of the paper, rather than having to scan the entire appendix (e.g. for Appendix B). For example, Line 33, the footnote should be moved to avoid confusion. Line 83, continuum armed vs continuous armed. Line 113, the ball should be noted as closed. There is a weird bold n on line 201. Some additional proofreading will likely illuminate these. Shrinkt in Fig.1, etc. While the proof technique used for showing upper bounds is algorithmic, it remains theoretical (this is not a detriment to the paper; just something to be noted). Due to time constraints, I have not gone through the entire proofs in detail, but as far as I can tell they appear to be reasonable. Some technical comments: Local Minimax rate definition (143-164): Expansion on point (1) on how the additional less restrictive assumptions are useful would improve this point. These might be shortened a bit in order (with a bit of expansion in the appendix) in order to provide some more technical proof sketch details in the body of the paper. For the intended audience, I feel that may be more useful. Example 1,2 (243,244): It seems like condition (A3) is the condition of proposition 2. Remark 1 (212): There is something missing in the last part of the sentence; the violations of the conditions can't be too severe in order for the argument to work. This may be worth expounding for another sentence or two. Theorem 1: It would be useful to give some sketch on the ingredients of the algorithm in the body of the paper at a high level.

Reviewer 3



This paper studies zero-order optimization with noisy observations. The authors study an "adaptive sampling" setting, where every query can depend on the results of previous queries. Although adaptive sampling gives the analyst an intuitive advantage over uniform sampling, the existing theoretical result doesn't confirm a more superior convergence rate for it. Through an information theoretic analysis, the paper shows that the classical rate is unimprovable even by active sampling. Further more, it presents a local convergence analysis, showing that if the true function lies in the neighborhood of some desirable reference function, then active sampling does enjoy a better convergence rate than uniform sampling. The gap is presented in both an upper bound and an almost matching lower bound. Overall, I enjoyed reading the paper. The technical results are solid, interesting and well-presented. The derivation of the lower bound for the active estimator involves some interesting technique. I have two minor questions: 1. The upper bound is presented in a way that with probability at most 1/4, then risk will be bounded by the c*R_n. Why don't the authors prove an upper bound on the expectation of the risk, or prove a high-probability upper bound? A in-expectation upper bound is apparently stronger. If a high-probability bound can be derived from the existing fixed probability bound by repeating the estimation multiple times, it is at least worth mentioning in the paper. 2. It looks like that many constants in the paper depend on the dimension d. For example, the constant C_0 can be exponential in d. Therefore the gap between the upper bound and the lower bound may also have this exponential dependence. If this is true, then it should be explicitly stated in the paper, because in real applications d can be large.